# Phylogenetic analyses of Norwegian *Tenacibaculum* strains confirm high bacterial diversity and suggest circulation of ubiquitous virulent strains

Erwan Lagadec[1]*, Sverre Bang Småge[2], Christiane Trösse[1], Are Nylund[1]

**1** Fish Disease Research Group, Department of Biological Sciences, University of Bergen, Bergen, Norway,
**2** Cermaq Group AS, Oslo, Norway

* erwan.lagadec@uib.no

**Data Availability Statement:** All relevant data are within the manuscript and its Supporting information files. All nucleotide sequences are available from the GenBank database (https://www.

## Abstract

Tenacibaculosis is a bacterial ulcerative disease affecting marine fish and represents a major threat to aquaculture worldwide. Its aetiological agents, bacteria belonging to the genus Tenacibaculum, have been present in Norway since at least the late 1980's and lead to regular ulcerative outbreaks and high mortalities in production of farmed salmonids. Studies have shown the presence of several Tenacibaculum species in Norway and a lack of clonality in outbreak-related strains, thus preventing the development of an effective vaccine. Hence, a thorough examination of the bacterial diversity in farmed fish presenting ulcers and the geographical distribution of the pathogens should provide important insights needed to strengthen preventive actions. In this study, we investigated the diversity of Tenacibaculum strains isolated in 28 outbreaks that occurred in Norwegian fish farms in the period 2017–2020. We found that 95% of the 66 strains isolated and characterized, using an existing MultiLocus Sequence Typing system, have not previously been identified, confirming the high diversity of this genus of bacteria in Norway. Several of these Tenacibaculum species seem to be present within restricted areas (e.g., Tenacibaculum dicentrarchi in western Norway), but phylogenetic analysis reveals that several of the strains responsible of ulcerative outbreaks were isolated from different localities (e.g., ST- 172 isolated from northern to southern parts of Norway) and/or from different hosts. Understanding their reservoirs and transmission pathways could help to address major challenges in connection with prophylactic measures and development of vaccines.

## Introduction

Tenacibaculosis, caused by members of the genus *Tenacibaculum*, is an ulcerative skin disease of many economically important marine farmed fish species worldwide. This disease is of major concern for global fish production and is currently considered one of the most threatening bacterial diseases impacting mariculture [1]. Tenacibaculosis is associated with

ncbi.nlm.nih.gov/genbank/). All accession numbers are listed in Supporting Information S2.

**Funding:** This study was supported by a FHF grant (Fiskeri og Havbruksnæringens Forskningsfond, LimiT project #901433). The grant was received by AN. The funders had no role in study design, data collection and analysis, decision to publish or preparation of the manuscript.

**Competing interests:** The authors have declared that no competing interests exist.

characteristic clinical signs such as ulcerative skin lesions, mouth erosion, frayed fins, and tail. The best-known pathogen in the genus is the type species *Tenacibaculum maritimum* which has been reported from disease outbreaks in farmed fish since the 1970s [2–7]. However, in recent years an increasing number of novel *Tenacibaculum* species have been isolated and associated with ulcerative skin disease in farmed marine fishes [8–14]. Tenacibaculosis has been reported from marine wild and farmed fish in Europe, Asia, North and South America and Australia [2, 4, 5, 9, 15–18].

The first time a *Tenacibaculum* species was associated with fish disease in Norway was in 1989 with the novel bacterium *Tenacibaculum ovolyticum*. This bacterium was isolated from the adherent epiflora of halibut eggs and was shown to be an opportunistic pathogen for halibut egg and larvae [9, 19]. Since then, an increasing number of novel *Tenacibaculum* spp. have been associated with ulcerative skin disease in, or shown to be pathogenic to, farmed marine fish in Norway [11, 14, 20–24]. In the recent years, the increased use of sea salts-containing Marine Agar (MA) for bacterial isolation has improved the recovery rate of *Tenacibaculum* spp. [19–21]. Thus, *Tenacibaculum* spp. have also been associated with the winter ulcer disease, previously attributed to the bacterium *Moritella viscosa* [25, 26]. From phylogenetic analyses it has been shown that there is a large *Tenacibaculum* spp. diversity in Norway [11, 14, 21, 23, 27, 28]. However, most *Tenacibaculum* isolates associated with ulcerative disease belong to *Tenacibaculum finnmarkense* genomovar *finnmarkense*, *Tenacibaculum finnmarkense* genomovar *ulcerans* and *Tenacibaculum dicentrarchi* [11, 23, 27].

Through a MultiLocus Sequence Analysis, the present study aimed at investigating the *Tenacibaculum* spp. diversity associated with outbreaks of ulcerative disease in Norwegian fish farms for the period 2017–2020 with an emphasis on the northern part, where the aquaculture industry suffers the most from tenacibaculosis. By expanding the analysis to *Tenacibaculum* spp. isolated all along the Norwegian coast, the presence of ubiquitous or multi-hosts *Tenacibaculum* strains that could be of importance in terms of future epidemic management of tenacibaculosis in salmon production was explored.

## Material and methods

### Bacteria isolation

Fish samples were collected from licensed Norwegian fish farms (Norwegian Directorate of Fisheries). The fish were treated by veterinarians and certified fish-health biologists according to the Norwegian Animal Welfare Act (01.01.2010) and the study strictly followed the regulations set by the Norwegian Food Safety Authority.

Farmed marine fishes (Atlantic herring (*Clupea harengus*), Atlantic halibut (*Hippoglossus hippoglossus*), Atlantic salmon (*Salmo salar*), lumpfish (*Cyclopterus lumpus)* and rainbow trout (*Oncorhynchus mykiss*)) suffering ulcerative skin disease in Norway were sampled in the period from 2017 to 2019. Fish were sampled either in the frame of a project surveying tenacibaculosis in farmed salmonids in Norway during ulcerative disease outbreaks (fish sampled by veterinarians or certified fish-health biologists on site during outbreaks) or in the frame of other research projects focusing neither on tenacibaculosis nor specifically on salmon (fish sampled by fish-health biologists from our laboratory). Bacteria were isolated from affected fresh tissues (i.e., gills, skin, mouth, fins and from internal organs) directly on site on Marine Agar (MA) (Difco 2216) as a routine medium used by fish veterinarians, or on blood marine agar (BAMA) and BAMA supplemented with kanamycin to a final concentration of 50 μg/ml in our laboratory. Although the isolation process was optimized for the detection of *Tenacibaculum* spp. (especially culture media), growth of other bacteria species that can potentially cause ulcers (e.g., *Moritella viscosa*) was monitored.

A total of 197 fish were examined and sampled. Bacteria isolated from field samples were cultured for a minimum of three days at 16 ˚C. Plates were visually checked for the presence of *Tenacibaculum* spp. by looking for characteristic colonies as described for *Tenacibaculum* spp. [7, 9, 19]. The colonies were further examined by using a light microscope to identify the long and slender rods typical of *Tenacibaculum* spp. cell morphology [19]. The β-haemolytic activity of the colonies was readily observed by looking at the plates through a light source. Colonies containing bacteria with *Tenacibaculum* spp. morphology were further sub-cultured onto BAMA and incubated for two days at 16 ˚C or 20 ˚C. Clonal *Tenacibaculum* spp. cultures were stored in 50:50 Biofreeze freezing medium (Biochrom™, Germany) and Marine broth (DIFCO 2216) in liquid nitrogen.

## DNA extraction, 16S rRNA PCR and sequencing

Genomic DNA was extracted from the recovered *Tenacibaculum* isolates either by using a DNeasy Blood & Tissue kit (Qiagen) or by heating single colonies in nuclease-free water at 96 ˚C for 5 min. The heated colonies were subsequently centrifuged at 10,000 G for 5 min and the DNA containing supernatant transferred into new vials. All genomic DNA were stored at—20 ˚C.

PCR and sequencing of the 16S rRNA gene were performed on all isolates using the primers 27F and 1518R [29]. Amplification details are given in [17]. PCR products were deposited on a 1% agarose electrophoresis gel stained with GelRed™ (Biotium, USA). Obtained amplicons were purified using ExoCleanUp FAST (VWR) in a Veriti thermal cycler (Applied Biosystems) at 37 ˚C for 5 min and 80 ˚C for 10 min before being sequenced at the Sequencing Facility at the University of Bergen (http://www.uib.no/seqlab) using the BigDye Terminator v3.1 Cycle Sequencing Kit (Applied Biosystems).

Consensus sequences from each sample were obtained using VectorNTI 9.0.0 software (Invitrogen) and a BLAST search [30] was performed for preliminary bacterial identification.

## Genetic characterization

Bacteria confirmed to be *Tenacibaculum* spp. from 16S rRNA gene sequence analyses were further included in a MLST analysis using primers targeting seven housekeeping genes *atpA* (567 bp), *dnaK* (573 bp), *glyA* (558 bp), *gyrB* (597 bp), *infB* (564 bp), *rlmN* (549 bp) and *tgt* (486 bp) as previously described [28].

The established *Tenacibaculum* MLST website (https://pubmlst.org/organisms/tenacibaculum-spp/) assigned unique allele identifiers for the seven loci considered and defined corresponding allelic profiles referred to as sequence types (STs) (S1 Table). In order to determine the DNA relatedness among *Tenacibaculum* strains, a minimum-spanning tree based on the seven loci was generated in PHYLOViZ 2.0 using the goeBURST Full MST algorithm (https://online.phyloviz.net/index) [31, 32].

MultiLocus Sequence Analysis (MLSA) was conducted to compare the *Tenacibaculum* spp. strains included in this study with strains previously described in other studies. The analysis included sequences of the following additional *Tenacibaculum* spp.: *Tenacibaculum finnmarkense* genomovar *finnmarkense* strain HFJ [14], and 36 sequences obtained from GenBank: 15 *Tenacibaculum* spp. strains isolated in Norway (TNO) and 21 type strains in the genus *Tenacibaculum* (*T. adriaticum*[T], *T. aestuarii*[T], *T. aiptasiae*[T], *T. amylolyticum*[T], *T. crassostreae*[T], *T. discolor*[T], *T. dicentrarchi*[T], *T. finnmarkense* genomovar *finnmarkense*[T], *T. finnmarkense* genomovar *ulcerans*[T], *T. gallaicum*[T], *T. geojense*[T], *T. jejuense*[T], *T. litopenaei*[T], *T. litoreum*[T], *T. lutimaris*[T], *T. maritimum*[T], *T. mesophilum*[T], *T. ovolyticum*[T], *T. piscium*[T], *T. skagerrakense*[T], *T.*

*soleae*[T]). In accordance with [28], *Kordia algicida* (Accession number NZ_DS544873.1) was used as an outgroup.

Sequence alignments were constructed for all seven loci separately using AlignX in the VectorNTI 9.0.0 software package (Invitrogen). The sequences were trimmed and adjusted to correct reading frames in GeneDoc [33]. Concatenation of the seven housekeeping genes was performed using Kakusan4 [34]. The substitution rate, codon position and best fit model for the individual loci were calculated with Kakusan4. The Bayesian phylogenetic analysis was performed in MrBayes 3.2 [35] with a proportional codonproportional data bloc and a Markov Chain Monte Carlo (MCMC) analysis. The run included 20,000,000 generations and trees were sampled every 1000 generations. The initial 10,000 trees were discarded as a conservative "burn-in" in TreeAnnotator and the final tree was visualized in FigTree v1.4.3 (http://tree.bio. ed.ac.uk/). GenBank accession numbers of the sequences from this study are provided in S2 Table.

## Results

### Isolation of bacteria from diseased fish

In the present study, more than 300 *Tenacibaculum* spp. isolates were recovered from five fish species: Atlantic herring (*Clupea harengus*), Atlantic halibut (*Hippoglossus hippoglossus*), Atlantic salmon (*Salmo salar*), lumpfish (*Cyclopterus lumpus)* and rainbow trout (*Oncorhynchus mykiss*). Bacterial colony characteristics are described in S3 Table. Twenty-eight outbreaks in fish farms were investigated in eight counties along the Norwegian coast (Akershus, Finnmark, Hordaland, Møre og Romsdal, Nordland, Rogaland, Troms and Sogn og Fjordane). Sampling details and tenacibaculosis ulcer pictures are given in S4 Table and S1 Fig, respectively. We successfully amplified and sequenced seven MLST loci from 66 strains isolated from the 28 outbreaks (Table 1).

To note, several other bacteria were isolated (e.g., *Vibrio* spp., *Pseudoalteromonas* spp., *Photobacterium* spp.), always isolated colonies outcompeted by the growth of a large number of *Tenacibaculum* spp. colonies.

### Genetic characterization

The *Tenacibaculum* MLST website assigned a total of 29 STs from the *Tenacibaculum* spp. isolates included in this study, with 27 being new STs to the database. Two STs were previously isolated from Atlantic salmon in 2010 (ST-52 and ST-53) and in 2011 (ST-52). In the present study, ST-53 was isolated from a diseased Atlantic salmon farmed in Troms in April 2018. ST-52 was isolated from two Atlantic salmon, in March 2019 in Akershus and in May 2019 in Troms (Fig 1).

Of the 27 new STs, 22 were isolated only once during 22 different outbreaks of tenacibaculosis. Five STs were isolated during more than one outbreak: ST-160, ST-162, ST-107, ST-152 and ST-172.

ST-160 and ST-162 were associated with outbreaks of ulcerative skin disease that occurred in April 2018 in Troms. Both were isolated from Atlantic salmon from the same farm during two samplings separated by seven days. ST-160 was isolated from both a skin ulcer and from the head kidney of the same fish. Two Atlantic salmon with skin ulcers were also infected with ST-160 in another farm in Troms during the same period (April 2018).

ST-107 was isolated from Atlantic salmon during tenacibaculosis outbreaks at a single site in Hordaland in March, April, and November 2019. ST-107 was also isolated from a single site rearing herring that presented with ulcerative disease in February 2020.

**Table 1.** Data associated with the 66 *Tenacibaculum* strains isolated during this study.

| ID | Sampling number | Country | Region | Date of collection | Fish host | Fish number | Organ isolation | Haemolytic activity |
|---|---|---|---|---|---|---|---|---|
| LIM001 | 1 | Norway | Sogn og F.* | Apr-2018 | *Salmo salar* | #1 | skin ulcer | yes |
| LIM002 | 1 | Norway | Sogn og F.* | Apr-2018 | *Salmo salar* | #2 | skin ulcer | yes |
| LIM003 | 1 | Norway | Sogn og F.* | Apr-2018 | *Salmo salar* | #4 | skin ulcer | no |
| LIM004 | 1 | Norway | Sogn og F.* | Apr-2018 | *Salmo salar* | #4 | skin ulcer | no |
| LIM005 | 2 | Norway | Troms | Apr-2018 | *Salmo salar* | #1 | skin ulcer | yes |
| LIM006 | 2 | Norway | Troms | Apr-2018 | *Salmo salar* | #1 | skin ulcer | no |
| LIM007 | 2 | Norway | Troms | Apr-2018 | *Salmo salar* | #2 | skin ulcer | yes |
| LIM008 | 2 | Norway | Troms | Apr-2018 | *Salmo salar* | #2 | skin ulcer | yes |
| LIM009 | 2 | Norway | Troms | Apr-2018 | *Salmo salar* | #8 | skin ulcer | yes |
| LIM010 | 3 | Norway | Troms | Apr-2018 | *Salmo salar* | #1 | skin ulcer | no |
| LIM011 | 5 | Norway | Troms | Sep-2019 | *Salmo salar* | #1 | skin ulcer | yes |
| LIM012 | 5 | Norway | Troms | Sep-2019 | *Salmo salar* | #2 | skin ulcer | no |
| LIM013 | 6 | Norway | Hordaland | Apr-2017 | *Salmo salar* | #2 | skin ulcer | yes |
| LIM014 | 6 | Norway | Hordaland | Apr-2017 | *Salmo salar* | #3 | skin ulcer | yes |
| LIM016 | 2 | Norway | Troms | Apr-2018 | *Salmo salar* | #3 | skin ulcer | no |
| LIM017 | 2 | Norway | Troms | Apr-2018 | *Salmo salar* | #7 | skin ulcer | no |
| LIM018 | 3 | Norway | Troms | Apr-2018 | *Salmo salar* | #4 | skin ulcer | yes |
| LIM020 | 4 | Norway | Troms | Apr-2018 | *Salmo salar* | #3 | skin ulcer | yes |
| LIM023 | 7 | Norway | Troms | Apr-2018 | *Salmo salar* | #4 | skin ulcer | yes |
| LIM024 | 8 | Norway | Hordaland | Mar-2019 | *Salmo salar* | #1 | skin ulcer | yes |
| LIM025 | 8 | Norway | Hordaland | Mar-2019 | *Salmo salar* | #2 | skin ulcer | yes |
| LIM026 | 9 | Norway | Nordland | Mar-2019 | *Salmo salar* | #1 | skin ulcer | yes |
| LIM027 | 9 | Norway | Nordland | Mar-2019 | *Salmo salar* | #2 | head ulcer | yes |
| LIM032 | 10 | Norway | Akershus | Mar-2019 | *Salmo salar* | #1 | gill | yes |
| LIM033 | 10 | Norway | Akershus | Mar-2019 | *Salmo salar* | #2 | gill | yes |
| LIM036 | 11 | Norway | Hordaland | Apr-2019 | *Salmo salar* | #1 | mouth, skin ulcer | yes |
| LIM040 | 12 | Norway | Nordland | Apr-2019 | *Salmo salar* | #6 | skin ulcer | yes |
| LIM042 | 13 | Norway | Hordaland | Apr-2019 | *Salmo salar* | #11 | mouth, gill | yes |
| LIM043 | 14 | Norway | Hordaland | Apr-2019 | *Salmo salar* | #17 | mouth, gill | yes |
| LIM044 | 15 | Norway | Troms | Apr-2019 | *Salmo salar* | #1 | mouth ulcer | yes |
| LIM046 | 2 | Norway | Troms | Apr-2018 | *Salmo salar* | #1 | kidney | yes |
| LIM047 | 2 | Norway | Troms | Apr-2018 | *Salmo salar* | #3 | kidney | no |
| LIM048 | 2 | Norway | Troms | Apr-2018 | *Salmo salar* | #9 | kidney | yes |
| LIM049 | 2 | Norway | Troms | Apr-2018 | *Salmo salar* | #3 | skin ulcer | no |
| LIM050 | 2 | Norway | Troms | Apr-2018 | *Salmo salar* | #4 | skin ulcer | yes |
| LIM051 | 2 | Norway | Troms | Apr-2018 | *Salmo salar* | #5 | skin ulcer | yes |
| LIM052 | 2 | Norway | Troms | Apr-2018 | *Salmo salar* | #6 | skin ulcer | yes |
| LIM053 | 2 | Norway | Troms | Apr-2018 | *Salmo salar* | #9 | skin ulcer | yes |
| LIM054 | 3 | Norway | Troms | Apr-2018 | *Salmo salar* | #2 | skin ulcer | no |
| LIM055 | 3 | Norway | Troms | Apr-2018 | *Salmo salar* | #3 | skin ulcer | no |
| LIM056 | 16 | Norway | Hordaland | Oct-2017 | *Salmo salar* | #1 | mouth | no |
| LIM057 | 16 | Norway | Hordaland | Oct-2017 | *Salmo salar* | #1 | mouth | no |
| LIM058 | 16 | Norway | Hordaland | Oct-2017 | *Salmo salar* | #3 | mouth | yes |
| LIM059 | 7 | Norway | Troms | Apr-2018 | *Salmo salar* | #3 | kidney | no |
| LIM060 | 7 | Norway | Troms | Apr-2018 | *Salmo salar* | #4 | skin ulcer | no |
| LIM061 | 6 | Norway | Hordaland | Apr-2017 | *Salmo salar* | #1 | mouth | yes |
| LIM062 | 11 | Norway | Hordaland | Apr-2019 | *Salmo salar* | #3 | mouth, gill | yes |

(*Continued*)

**Table 1.** (Continued)

| ID | Sampling number | Country | Region | Date of collection | Fish host | Fish number | Organ isolation | Haemolytic activity |
|---|---|---|---|---|---|---|---|---|
| LIM063 | 17 | Norway | Hordaland | Nov-2019 | *Salmo salar* | #1 | skin ulcer | yes |
| LIM064 | 18 | Norway | Rogaland | Mar-2019 | *Hippoglossus hippoglossus* | #1 | skin ulcer | yes |
| LIM065 | 19 | Norway | Rogaland | Apr-2019 | *Hippoglossus hippoglossus* | #1 | skin ulcer | yes |
| LIM066 | 20 | Norway | Hordaland | Apr-2019 | *Cyclopterus lumpus* | #1 | skin ulcer, eye | no |
| LIM067 | 21 | Norway | Troms | Mar-2019 | *Salmo salar* | #4 | skin ulcer | no |
| LIM068 | 21 | Norway | Troms | Mar-2019 | *Salmo salar* | #1 | skin ulcer | no |
| LIM069 | 14 | Norway | Hordaland | Apr-2019 | *Salmo salar* | #23 | skin ulcer | no |
| LIM070 | 13 | Norway | Hordaland | Apr-2019 | *Salmo salar* | #26 | mouth, gill | yes |
| LIM071 | 22 | Norway | Hordaland | Mar-2019 | *Salmo salar* | #24 | mouth, gill | yes |
| LIM072 | 23 | Norway | Sogn og F.* | Dec-2018 | *Cyclopterus lumpus* | #1 | kidney | no |
| LIM073 | 23 | Norway | Sogn og F.* | Dec-2018 | *Cyclopterus lumpus* | #1 | skin ulcer | yes |
| LIM074 | 24 | Norway | Rogaland | Feb-2019 | *Oncorhynchus mykiss* | #1 | skin ulcer | yes |
| LIM075 | 24 | Norway | Rogaland | Feb-2019 | *Oncorhynchus mykiss* | #2 | skin ulcer | no |
| LIM076 | 25 | Norway | Troms | May-2019 | *Salmo salar* | #1 | mouth | yes |
| LIM077 | 25 | Norway | Troms | May-2019 | *Salmo salar* | #3 | mouth | yes |
| LIM078 | 26 | Norway | Rogaland | Jul-2019 | *Hippoglossus hippoglossus* | #1 | gill | no |
| LIM079 | 27 | Norway | Rogaland | Oct-2019 | *Hippoglossus hippoglossus* | #1 | gill | no |
| LIM080 | 28 | Norway | Hordaland | Feb-2020 | *Clupea harengus* | #1 | skin ulcer | yes |
| LIM081 | 28 | Norway | Hordaland | Feb-2020 | *Clupea harengus* | #2 | skin ulcer | yes |

*: Sogn og Fjordane.

ST-152 was isolated from Atlantic salmon in Hordaland (April 2017, March, and April 2019), in Troms (April and September 2018) and in Akershus (March 2019).

The most common ST found in this dataset was ST-172. It was isolated from salmon in three different counties: Sogn og Fjordane (April 2018), Nordland (Mars and April 2019) and Troms (April 2018, April, and May 2019). ST-172 was also isolated from rainbow trout (February 2019) and lumpfish in Rogaland (April 2019). This ST was associated with severe tenacibaculosis.

## MultiLocus Sequence Analysis (MLSA)

A phylogenetic tree was constructed using the 66 concatenated sequences (3894 bp) included in this study. In addition, the analysis included the strain HFJ [14], 15 *Tenacibaculum* spp. strains previously isolated in Norway [28], and 21 type strains in the *Tenacibaculum* genus (Fig 2).

Fifty-seven out of the 66 strains (83%) belong to a large and highly diversified *T. dicentrarchi/T. finnmarkense* group of isolates that could be further separated into five highly supported distinct clades (clades I, II, III, IV and V).

Clade I (the *T. finnmarkense* genomovar *ulcerans* clade) consists of 14 Norwegian *Tenacibaculum* strains including strains isolated from Atlantic salmon from 1996 to 2011 (n = 5) and Atlantic cod in 2010 (n = 3) [20, 23, 28]. The six strains from this study were isolated from Atlantic salmon (n = 5) and lumpfish (n = 1). None of these strains showed β-haemolytic activity when grown on BAMA.

Clade II (the *T. dicentrarchi* clade) consists of eight isolates and includes the *T. dicentrarchi* type strain. The six isolates from the present study were isolated from Atlantic salmon in March (LIM024 and LIM025), April (LIM036) and November 2019 (LIM063) and from

A - Regions

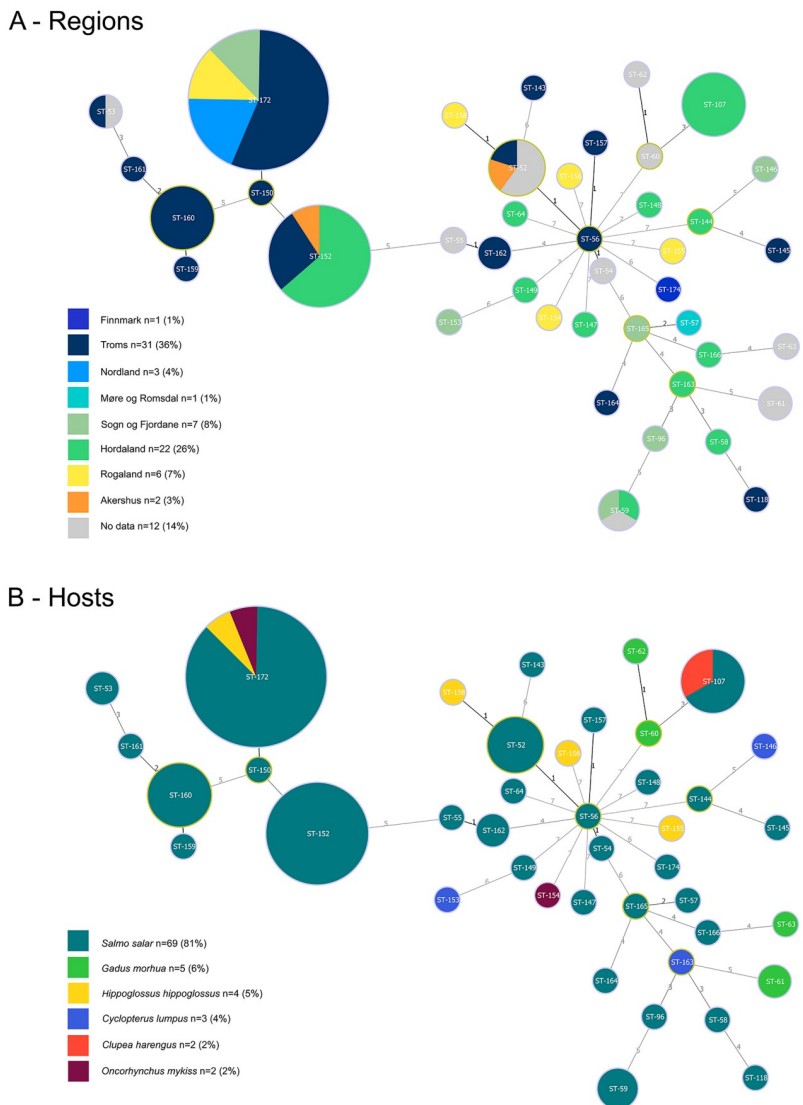

B - Hosts

**Fig 1. Minimum spanning tree.** Minimum spanning tree of *Tenacibaculum* strains based on 7 loci (*atpA*, *dnaK*, *glyA*, *gyrB*, *infB*, *rlmN* and *tgt*) using the goeBURST Full MLST algorithm. The sequence types (STs) are identified by a number (ST-, see https://pubmlst.org/tenacibaculum/). The circle size reflects their abundance in the data set. Group founders are indicated by a yellow circle. The analysis includes 67 new isolates for the database (66 STs from this study and ST-174 assigned to HFJ strain), and 18 STs already published. The DNA relatedness is presented with emphasis on regions (A) or hosts (B).

Atlantic herring in February 2020 (LIM080 and LIM081). All six isolates presented β-haemolytic activity on blood agar and were associated with clinical signs of tenacibaculosis. The clade II isolates were only isolated in Hordaland and showed 100% identity based on the seven housekeeping genes (ST-107). The *Tenacibaculum* isolates TNO012 and TNO015 were recovered from Atlantic cod in 2009 and 2010 and are genetically distinct from each other and from all other LIM isolates in the current study.

Clade III (the *T. finnmarkense* genomovar *finnmarkense* clade) includes the HFJ strain, 44 isolates recovered during this study and seven isolates from Atlantic salmon sampled in 2010 and 2011 [28]. Forty-two isolates were collected from Atlantic salmon in Akershus, Finnmark,

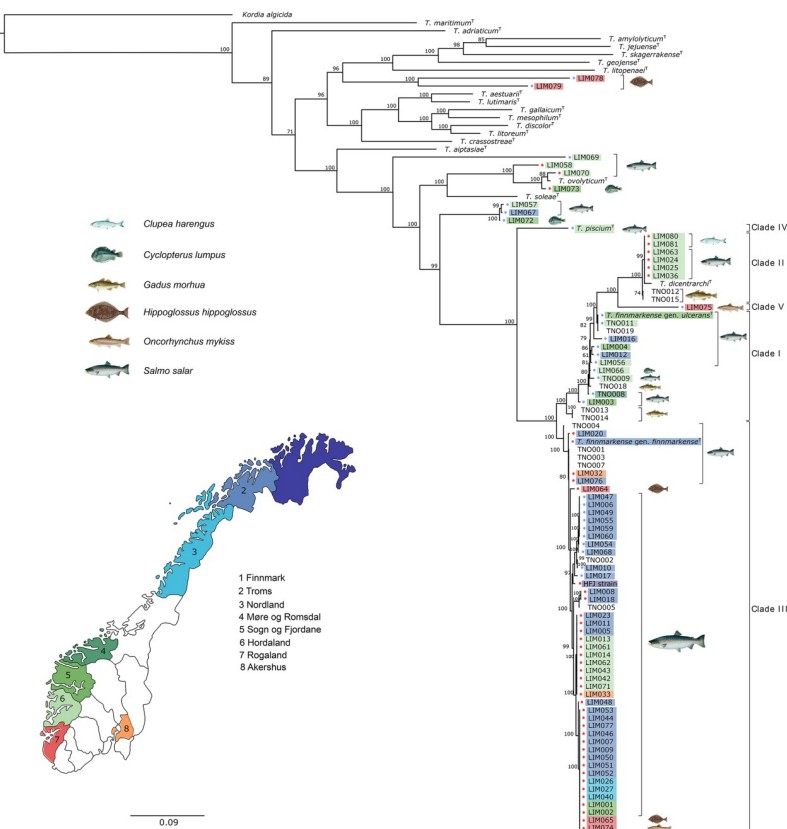

**Fig 2. Bayesian analysis.** Bayesian phylogenetic tree of 103 *Tenacibaculum* strains based on concatenated sequences of seven genes (*atpA*, *dnaK*, *glyA*, *gyrB*, *infB*, *rlmN* and *tgt*, total size: 3894 bp). The data set includes 66 *Tenacibaculum* strains isolated during this study (LIM strains), the HFJ strain previously isolated and 36 strains from GenBank (15 from [28] (TNO strains) and 21 *Tenacibaculum* Type Strains). *Kordia algicida* was used as an outgroup. Posterior probabilities are indicated at each node. Fish symbols indicate the host from which each strain has been isolated (LIM and TNO) and colours indicate the Norwegian administrative regions, accordingly to the map. Colour dots indicate if the strains show haemolytic activity on blood agar (red dot), not (blue dot) or variable (red/blue dot). Clades I-V represent the structure of the *T. finnmarkense/T. dicentrarchi* group of isolates. The Norwegian map was reprinted from https://doi.org/10.1371/journal.pone.0215478 under a CC BY license, with permission from Are Nylund, original copyright 2019. This map is similar but not identical to the original image and is therefore for illustrative purposes only.

Hordaland, Nordland and Troms. Three isolates were found in rainbow trout (LIM074) and Atlantic halibut (LIM064 and LIM065) in Rogaland. All these isolates showed β-haemolytic activity on blood agar except for ten isolates (TNO002, LIM006-010-017-047-049-054-055-059-060-068).

Clade IV and clade V are both represented by a single isolate; *T. piscium* type strain (isolated from an Atlantic salmon in 1998) and LIM075 (isolated from a rainbow trout in Rogaland in February 2019), respectively. Like *T. piscium* [11], LIM075 did not present any β-haemolytic activity on blood agar.

LIM057, LIM067 and LIM072 form a monophyletic group. They were isolated in Hordaland (LIM057 in October 2017), Sogn og Fjordane (LIM072 in December 2018) and Troms (LIM067 in March 2019), from 2 different hosts: Atlantic salmon (LIM057 and LIM067) and lumpfish (LIM072). None of them showed any β-haemolytic activity on blood agar.

Three isolates closely related to *T. ovolyticum*. LIM058 and LIM070 were isolated from salmon in Hordaland in October 2017 and April 2019, respectively. LIM073 was isolated from a lumpfish in Sogn og Fjordane in December 2018. They all showed β-haemolytic activity on blood agar.

LIM069 was isolated from an Atlantic salmon skin ulcer in Hordaland in April 2019. It did not present any β-haemolytic activity on blood agar.

LIM078 and LIM079 are two distinct *Tenacibaculum* spp. strains isolated from farmed Atlantic halibut in Rogaland. Neither strain showed β-haemolytic activity on blood agar.

## Discussion

The diagnosis of tenacibaculosis has usually been based on histology and isolation of *Tenacibaculum* spp. from sick fish followed by morphological, biochemical, and serological characterization of the bacteria and analysis of their antimicrobial susceptibility profiles [8]. MultiLocus Sequence Analysis (MLSA) has, however, been increasingly used for bacterial genotyping [17, 23, 28, 36, 37] and has become an important tool for diagnosis and studies of epizootics. The results from the current study confirm the large diversity of *Tenacibaculum* spp. strains present in fish farms along the Norwegian coast and their association with ulcerative disease outbreaks.

Although the isolation process was optimized for the detection of *Tenacibaculum* spp., the implication of other bacteria species likely to be the cause of the observed ulcers can be fairly dismissed.

### High diversity of Norwegian *Tenacibaculum* spp. strains

This study presents the genotyping of 66 *Tenacibaculum* strains isolated from April 2017 to February 2020 in Norway. Phylogenetic analyses show that 57 strains belong to a major lineage structured into five different clades consisting of *T. finnmarkense* genomovar *ulcerans* Clade I, *T. dicentrarchi* Clade II, *T. finnmarkense* genomovar *finnmarkense* Clade III, *Tenacibaculum piscium* Clade IV, and a putative new species Clade V (Fig 2). This is in accordance with previous phylogenetic analyses of *Tenacibaculum* spp. isolated from farmed fishes in Norway [11, 23, 27]. Several strains are not closely related to any *Tenacibaculum* spp. type strains (LIM078 and LIM079 isolated from Atlantic halibut in Rogaland, LIM069 isolated from Atlantic salmon in Hordaland; LIM057, LIM067 and LIM072 isolated from Atlantic salmon and lumpfish in Troms, Hordaland, and Sogn og Fjordane). Nevertheless, further identification steps must be undertaken to conclude whether these new strains represent novel *Tenacibaculum* species.

Altogether, these results confirm the large *Tenacibaculum* diversity in Norway, and suggest that other strains have yet to be identified.

The phylogenetic analysis clearly showed that most of the isolates recovered from Atlantic salmon suffering tenacibaculosis belong to the species *T. finnmarkense* genomovar *finnmarkense* and have β-haemolytic activity (except for five STs clustering in a strongly supported clade). The *T. dicentrarchi* strain isolated in Hordaland (ST-107, β-haemolytic) was associated with tenacibaculosis and mortality in both Atlantic salmon and Atlantic herring. The pathogenicity of *T. dicentrarchi* strains has been already described in several fish species including European sea bass, wrasse, Atlantic cod, and Atlantic salmon from Chile [13, 28, 37]. Olsen et al. suggested that three of the four *T. dicentrarchi* strains isolated from asymptomatic salmon could have a higher pathogenicity towards non-salmonids species [23]. The severe symptoms associated with ST-107 (LIM 024-025-036-063-080-081) suggests pathogenicity towards Atlantic salmon and Atlantic herrings and indicate that strains pathogenic to salmon may also occur in Norway. The three strains closely related to *T. ovolyticum* were associated with ulcers and

showed β-haemolytic activities when grown on blood agar plates. Even though *T. ovolyticum* is already known for its ability to dissolve halibut eggshell, which could lead to the death of the embryo [9, 38], it is to our knowledge the first report of pathogenicity of *T. ovolyticum* toward Atlantic salmon and lumpfish. Nevertheless, it is uncommon and therefore unlikely to be a major threat to aquaculture.

β-haemolytic activity is commonly associated with pathogenic bacteria [39]. Among the strains isolated in the context of this study, non-haemolytic *Tenacibaculum* strains seem to be less pathogenic and therefore may not lead to tenacibaculosis outbreaks. Indeed, when associated with outbreaks, Clade I strains were isolated together with other haemolytic strains (LIM003, LIM004, LIM012, LIM016). Some Clade I strains were sporadically associated with ulcer symptoms (LIM066 for instance). This reinforces the need for the use of blood-supplemented culture media in the field. Estimating as early as possible the virulence of a strain is necessary in order to implement rapid sanitary measures. Nevertheless, it is important to temper this assessment, as non-haemolytic strains can also prove to be pathogenic. For instance, the non-haemolytic ST-160 (Clade III) was also associated with tenacibaculosis. While presence of haemolytic activity is to be seriously considered, all strains should be monitored.

## Presence of clinically relevant ubiquitous strains

*Tenacibaculum finnmarkense* isolates from clades I and III are present all along the Norwegian coast, from Akershus in the southeast to Finnmark in the north. It is interesting to note that all the STs from Clade III except one (LIM064, isolated in Rogaland) have been isolated at least once in Troms or Finnmark. To note, *Tenacibaculum finnmarkense* has also been isolated outside of Norway, from Chilean Atlantic salmon, coho salmon, and rainbow trout (*Oncorhynchus kisutch*) [27, 36, 40]. *Tenacibaculum dicentrarchi strains* and the strains closely related to *T. ovolyticum* were only isolated in the western part of Norway. The reason why *T. dicentrarchi* and *T. ovolyticum* seem to be mostly present in western Norway is unknown but may be linked to several factors such as higher seawater temperature in western Norway compared to northern Norway.

Ninety-three % of the STs described herein were new to the MLST database. Only two STs (ST-52 and ST-53) were previously isolated from Atlantic salmon in 2010 and 2011 [23]. The majority of these new STs (81%) were recovered only once and therefore each were associated with a single locality and a single host species. This is in accordance with previous results that showed that the overall lack of clonality and host specificity among the Norwegian *Tenacibaculum* isolates indicated that tenacibaculosis infections arise as local epidemics involving multiple strains [23]. However, several STs were isolated from different localities and/or from different hosts and are therefore of extreme importance for the aquaculture. ST-152 is associated with tenacibaculosis outbreaks in farmed Atlantic salmon in three different counties: Troms, Hordaland and Akershus. ST-172 is the strain with the widest distribution, from Troms in the north, to Rogaland in the south. This ST, always associated with severe tenacibaculosis, might have a wider tolerance to different seawater conditions (salinity and temperature) than *T. dicentrarchi*. More importantly, both STs were isolated throughout the sampling period, from April 2017 to May 2019. We isolated ST-172 again in March 2020 during an outbreak in an Atlantic salmon farm in Troms. ST-172 has been isolated from three different fish species: Atlantic salmon, rainbow trout and lumpfishes. Even though it is uncommon, the ability of some *Tenacibaculum* strains to infect several fish species has been shown already [23, 28], but they were limited to a restricted geographic area. Likewise, the *T. dicentrarchi* related ST-107 isolated from both Atlantic salmon and Atlantic herring, was found only in Hordaland. But ST-172 and ST-152 present the ability to spread within a broad geographic area. How they

spread out along the Norwegian coast and how they maintain themselves in the environment between outbreaks remain unknown, but several hypotheses can be made. Avendaño-Herrera et al. suggested that seawater is not an important route of transmission for *T. maritimum* and hypothesised that bacteria need to be attached to particle or substrate such as sediments or fish mucus to be maintained until favourable conditions occur [41]. Levipan et al. showed that all the *T. dicentrarchi* strains in their study were able to adhere to and form biofilms on polystyrene surfaces [42]. Survival of *T. maritimum* in fish mucus [43] suggests that bacteria can colonize and spread together with its hosts. Fish movements at different life stages can therefore be a spreading route for their associated bacteria. Furthermore, the transport of pathogens by fish ectoparasites have already been reported. For instance, the sea lice *Lepeophtheirus salmonis*, a common salmonid associated ectoparasite, can carry several micro-organisms such as *Aeromonas salmonicida* [44] and *T. maritimum* [45]. Thus, studies assessing the transport of *Tenacibaculum* sp. need to be implemented.

## Conclusion

MultiLocus Sequence Typing is a powerful tool for studies of epizootics caused by bacteria and other pathogens. It is highly efficient to get an overview of a bacterial community and to separate strains with putative different pathogenicity. Overall, the results from this study consolidate the conclusions of previous studies and confirm the high diversity of the *Tenacibaculum* spp. strains infecting farmed fishes in Norway. This high diversity of *Tenacibaculum* spp. strains and their broad geographical distribution is a major challenge for aquaculture, not only in Norway but also globally. Attempts to tackle some of these major outbreaks by focusing on vaccines have so far been unsuccessful, which increases the importance of knowledge about reservoirs, transmission routes and virulence differences.

## Supporting information

**S1 Table. Allelic profiles.** Sequence type (ST) for 67 *Tenacibaculum* strains and unique allelic identifiers for the seven loci considered in the MLST scheme assigned by the established *Tenacibaculum* MLST website (https://pubmlst.org/tenacibaculum/).
(DOCX)

**S2 Table. GenBank accession numbers of the sequences produced in this study.**
(DOCX)

**S3 Table. Colony characteristics.** Growth characteristics on blood Marine Agar of the 29 sequence types isolated in the study.
(DOCX)

**S4 Table. Sampling details.**
(XLSX)

**S1 Fig. Details on tenacibaculosis ulcers.** A: sampling 1. B: sampling 9. C: sampling 11. D: sampling 12. E: sampling 14. F: sampling 15. G: sampling 21. H: sampling 27. I: sampling 28. Image credits: Photo by Erwan Lagadec (C, I), MarinHelse AS (B, F, G), Heidrun Nylund (A, D, E), Kjetil Solheim (H).
(TIF)

## Acknowledgments

We are deeply thankful to all the farmers and veterinarians from the Norwegian fish farms who greatly helped in the sampling. We also want to thank Lars Are Hamre (Sea Lice Research Centre, University of Bergen) and Kjetil Solheim (Fish health manager of Sterling White Halibut AS) who helped to obtain Atlantic salmon from Hordaland and Atlantic halibuts from Rogaland, respectively.

## Author Contributions

**Conceptualization:** Are Nylund.

**Data curation:** Erwan Lagadec, Sverre Bang Småge, Christiane Trösse, Are Nylund.

**Formal analysis:** Erwan Lagadec.

**Funding acquisition:** Are Nylund.

**Methodology:** Erwan Lagadec, Are Nylund.

**Project administration:** Are Nylund.

**Resources:** Are Nylund.

**Supervision:** Are Nylund.

**Validation:** Are Nylund.

**Writing – original draft:** Erwan Lagadec, Sverre Bang Småge, Are Nylund.

**Writing – review & editing:** Erwan Lagadec, Sverre Bang Småge, Christiane Trösse, Are Nylund.

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
