## [Decision Letter · Decision Letter 0]

11 Aug 2021

PONE-D-21-21158

Phylogenetic analyses of Norwegian Tenacibaculum strains confirm high bacterial diversity and suggest circulation of ubiquitous virulent strains

PLOS ONE

Dear Dr. Lagadec,

Thank you for submitting your manuscript to PLOS ONE. After careful consideration, we feel that it has merit but does not fully meet PLOS ONE’s publication criteria as it currently stands. Therefore, we invite you to submit a revised version of the manuscript that addresses the points raised during the review process.

Please see my comments below.

We look forward to receiving your revised manuscript.

Kind regards,

Lloyd Vaughan, PhD, BSc (Hons)

Academic Editor

PLOS ONE

Journal Requirements:

3. We note that Figure 2 in your submission contain map images which may be copyrighted. All PLOS content is published under the Creative Commons Attribution License (CC BY 4.0), which means that the manuscript, images, and Supporting Information files will be freely available online, and any third party is permitted to access, download, copy, distribute, and use these materials in any way, even commercially, with proper attribution. For these reasons, we cannot publish previously copyrighted maps or satellite images created using proprietary data, such as Google software (Google Maps, Street View, and Earth). For more information, see our copyright guidelines: http://journals.plos.org/plosone/s/licenses-and-copyright.

a) You may seek permission from the original copyright holder of Figure 2 to publish the content specifically under the CC BY 4.0 license.  

Additional Editor Comments:

Dear Dr. Lagadec,

Thankyou for submitting this interesting manuscript to PloSOne. As you can see from the reviewers' comments, they are all in favour of publishing your findings, as I am as well, but have reservations which would need to first be addressed. These are all experienced fish pathologists whose opinions I value highly and I was especially happy that they all consented to review your manuscript. That they did so is no doubt also a reflection of your own work and that of your group.

I very much agree with their suggestions and recommendations and am convinced that by addressing the points they have raised that this will improve your manuscript, and inevitably counter any likely queries which would have been raised by a critical readership.

The diversity in Tenacibaculum strains you and your colleagues have discovered in Norwegian waters is truly impressive and I have no doubt that groups in other parts of the world will be stimulated by your findings to explore their own regions in greater detail. This will be key to developing vaccines which can be widely applied to countering this threat to aquaculture and the ensuing consequences on the environment. In this context, the additional details and information sought by the reviewers will be invaluable for those planning such studies.

I am very much looking forward to your response to these suggestions, just as much as I enjoyed reading the manuscript you sent in.

Kind regards,

Lloyd Vaughan

Reviewers' comments:

Reviewer's Responses to Questions

**Comments to the Author**

1. Is the manuscript technically sound, and do the data support the conclusions?

Reviewer #1: Partly

Reviewer #2: Partly

Reviewer #3: Yes

Reviewer #4: Yes

2. Has the statistical analysis been performed appropriately and rigorously? 

Reviewer #1: Yes

Reviewer #2: Yes

Reviewer #3: N/A

Reviewer #4: Yes

3. Have the authors made all data underlying the findings in their manuscript fully available?

Reviewer #1: No

Reviewer #2: Yes

Reviewer #3: Yes

Reviewer #4: Yes

4. Is the manuscript presented in an intelligible fashion and written in standard English?

Reviewer #1: Yes

Reviewer #2: Yes

Reviewer #3: Yes

Reviewer #4: Yes

5. Review Comments to the Author

Reviewer #1: To better understanding the pathogen diversity underlying fish diseases is an important prerequisite for the control of the disease. The submitted Ms aims to examine the diversity of Tenacibaculum spp. Bacteria in Norwegian aquaculture. These bacteria are opportunistic pathogens and can cause ulcerative diseases. While originally, focus was T. maritimum , more recent research unraveled that there is much more diversity of Tenacibaculum species in aquaculture. The present study further extends the knowledge on the diversity of this genus, and as such it is of relevance.

Methodologically, the authors collected bacteria from marine fish farms suffering outbreaks of ulcerative skin disease during 2017-2019. The bacteria were isolated from diseased tissues, were cultured and then the 16S rRNA was sequenced using MLST. Personally, I am not an expert on MLST and thus are not in a position to judge on the adequacy of the molecular methodology. However, I have several questions on the sampling methodology: how many fish were sampled per farm? Were only diseased or also healthy fish/tissues sampled? Were all isolates from one fish pooled, were isolates from different individuals pooled, or were all isolates treated separately? Were only Tenacibuculum bacteria analysed or was a broader microbial characterization of the diseased fish performed? In particular I would be interested to know what evidence can be provided by the authors that the Tenacibaculum bacteria are indeed causative to the ulcerative changes. The mere presence of a bacterial species on a diseased fish dos not necessarily mean that this species is the etiological agent.

Results:

The data based in figure 1 are based on the analysis of 7 loci – what criteria were used to select those 7 loci?

Line 163: Since sampling details are inadequately described (see above), it is difficult to follow the results. For instance, in line 165 its says that ST-160 was isolated form both the skin ulcer and the head kidney of the same fish. Was ST-160 found in other tissues of this fish as well, or were other tissues not analysed? And was this fish the only fish with ST-160 in the farm/the only fish with ulcer, or were more fish positive for ST-160, and did they all show skin ulcer? This is a lot of detail, but this detail is critical to understand the association of T. spp with ulcer.

In addition, while the Results provide an in-depth MLST analysis, I missed an "epidemiological" analysis: For instance, the authors investigated a range of fish species: were certain strains/ST associated with certain host species or was no pattern detectable? Was there an association between farm conditions, ulcer prevalence/severity and presence of specific strains? If this study should go beyond a mere analysis of genetic diversity but would like to promote understanding of the disease epidemiology, more emphasis on this type of questions is required. In the way as presented now, the ms unfortunately does not provide insight how the results of this study would support "future epidemic management" (line 64)

The concluding sentence of the abstract says: "Understanding their reservoirs and transmission pathways could help to address major challenges in connection with prophylactic measures and development of vaccines". I fully agree with this statement, but I do not see how this Ms as presented now would support these aims. Interestingly, the authors address these questions in the Discussion, but unfortunately they do not apply it to their own study.

Reviewer #2: Phylogenetic analyses of Norwegian Tenacibaculum strains confirm high bacterial diversity and suggest circulation of ubiquitous virulent strains, by Lagadec et al.

In this manuscript, the authors provide useful information about the diversity of Tenacibaculum spp. in the Norwegian fish farms. Bacteria belonging to the Tenacibaculum genus have become important opportunistic pathogens for many different marine fishes and lately have been associated with ulcerative diseases in farmed salmon. The authors have collected several strains from different regions in Norway and have used MLST analysis to map their genetic diversity.

Although the manuscript is well-written, and the information provided is useful there are some important points that should be considered. It seems to me that the strains used in the study have been collected by fish vets during routine diagnostic analyses and are not part of a specific research survey of the team. This if fully acceptable and understandable, however if this is true, it should be specifically mentioned in the manuscript. I am not very sure if the media used in the isolation would favor the growth of all possible Tenacibaculum spp. These bacteria are difficult to isolate, and in many cases, there is overgrowth by other faster growing opportunistic bacteria. The use of general isolation media may have hindered the growth of other species like Tenacibaculum maritimum.

Furthermore, I am not convinced that the heamolytic activity of Tenacibaculum strains is a valid indicator of pathogenicity. There are several non-haemolytic bacterial pathogens which are highly virulent. Although the authors state that the occurrence of beta haemolysis is an indicator of pathogenicity (supported by a relevant citation), the non-occurrence does not mean that the strains are less virulent. Haemolytic activity has been used extensively by the authors, however how it was assessed is not described in the M&M. More care should be given in the use of haemolysis as virulence factor for Tenacibaculum in order to be convincing.

Specific comments to the authors:

L53: move (MA) after Marine Agar

L53: Explain how the use of MA associates Tenacibaculum with winter ulcers

L68: Mention how many samples were used

L75-78: Are these the best media for isolating Tenacibaculum spp? What about FMA or TMA? It seems that these media are mostly used for routine microbiology by the fish vets. If this is the case, it should be stated either in the M&M or in the introduction, since this is a limitation of the study that may have biased the results. If this is not the case, then you should provide references supporting the use of these media for the specific isolation of Tenacibaculum spp.

L:78 It is not clear later in the results how many of the isolates were Tenacibaculum. If there were other bacteria present, which were these? What is the prevalence of Tenacibaculum during the outbreaks?

L80: What are the characteristics of the colonies? are these the same in all media? were they described also in Blood Agar? In the papers cited the authors have used different media.

L82: Describe the morphology

L143: Table legend: You talk about haemolytic activity but how this was assessed is not in the M&M. Since haemolytic capacity of the bacteria plays an important role in your paper, the method used to assess it should be described

L190-191: How do you define “variable”? Have you used replicates in the heamolytic activity assay?

L266: What are the severe symptoms? Where were they described?

L276: How did you come to this conclusion that the non-haemolytic strains are less pathogenic? If the conclusion is based on the field data (eg morbidity/mortality), these data should be provided. Even then, the conclusion is really based on circumstantial evidence, and it is indicative if not speculative. Unless there are comparative data from challenge tests (if yes, then you should provide these data).

L281: The association of hemolysis to virulence or to be more accurate, the non-hemolysis to non- or reduced virulence is not supported by evidence and it is speculative.

Reviewer #3: This manuscript provides additional information on the epidemiology of Tenacibaculosis and should be published. The information provided is sound and useful but not a significantly novel contribution to fish bacteriology, but should be published as it provides good information on the distribution and epidemiology of the bacterial isolates of concern.

Reviewer #4: The manuscript submitted by Lagadec et al (PONE-D-21-21158) is a well presented and informative paper showing the diversity that exists amongst Tenacibaculum spp. strains in Norwegian aquaculture.

I have a few minor points for the authors to address.

Line 39 - bacterial diseases

Line 44 - Tenacibaculosis has been reported in marine wild and farmed fish in Europe, Asia,

Line 52 - Due to the increased use of Marine Agar (MA) for bacterial isolation in recent years, Tenacibaculum spp. have also been associated with the winter ulcer disease, previously attributed to the bacterium Moritella viscosa [25, 26].

Line 89 - 10,000 rpm – state xg rather than rpm

Lines 139/194/213/294 - Numbers at the start of sentences should be written in words rather than as a number.

Line 151- March 2019

Line 161 - Of the 27 new STs, 22 were isolated during a single outbreak of tenacibaculosis. Was the outbreak at a single site - mention the location of this outbreak?

Line 278 - Overall, non-haemolytic Tenacibaculum strains seem to be less pathogenic and may not lead to tenacibaculosis outbreaks. Can you state the evidence for this e.g. a reference or findings from your study?

6. PLOS authors have the option to publish the peer review history of their article (what does this mean?). If published, this will include your full peer review and any attached files.

Reviewer #1: No

Reviewer #2: No

Reviewer #3: No

Reviewer #4: No

---

## [Author Response · Author response to Decision Letter 0]

30 Sep 2021

PONE-D-21-21158

Phylogenetic analyses of Norwegian Tenacibaculum strains confirm high

bacterial diversity and suggest circulation of ubiquitous virulent strains

PLOS ONE

Dear Dr. Lagadec,

Thank you for submitting your manuscript to PLOS ONE. After careful

consideration, we feel that it has merit but does not fully meet PLOS ONE's

publication criteria as it currently stands. Therefore, we invite you to

submit a revised version of the manuscript that addresses the points raised

during the review process.

Please see my comments below.

Please submit your revised manuscript by Sep 25, 2021, 11:59PM. If you will

need more time than this to complete your revisions, please reply to this

message or contact the journal office at plosone@plos.org. When you're

ready to submit your revision, log on to

https://www.editorialmanager.com/pone/ and select the 'Submissions Needing

Revision' folder to locate your manuscript file.

 * A rebuttal letter that responds to each point raised by the academic

editor and reviewer(s). You should upload this letter as a separate file

labeled 'Response to Reviewers'.

 * A marked-up copy of your manuscript that highlights changes made to the

original version. You should upload this as a separate file labeled

'Revised Manuscript with Track Changes'.

 * An unmarked version of your revised paper without tracked changes. You

should upload this as a separate file labeled 'Manuscript'.

If you would like to make changes to your financial disclosure, please

include your updated statement in your cover letter. Guidelines for

resubmitting your figure files are available below the reviewer comments at

the end of this letter.

If applicable, we recommend that you deposit your laboratory protocols in

protocols.io to enhance the reproducibility of your results. Protocols.io

assigns your protocol its own identifier (DOI) so that it can be cited

independently in the future. For instructions see:

http://journals.plos.org/plosone/s/submission-guidelines#loc-laboratory-protocols.

Additionally, PLOS ONE offers an option for publishing peer-reviewed Lab

Protocol articles, which describe protocols hosted on protocols.io. Read

more information on sharing protocols at

https://plos.org/protocols?utm_medium=editorial-email&utm_source=authorletters&utm_campaign=protocols

[1].

We look forward to receiving your revised manuscript.

Kind regards,

Lloyd Vaughan, PhD, BSc (Hons)

Academic Editor

PLOS ONE

Journal Requirements:

When submitting your revision, we need you to address these additional

requirements.

1. Please ensure that your manuscript meets PLOS ONE's style requirements,

including those for file naming. The PLOS ONE style templates can be found

at

and

The authors: We have checked the PLOS ONE’s style requirements and verified that all documents followed the submission guidelines.

2. We note that you have stated that you will provide repository

information for your data at acceptance. Should your manuscript be accepted

for publication, we will hold it until you provide the relevant accession

numbers or DOIs necessary to access your data. If you wish to make changes

to your Data Availability statement, please describe these changes in your

cover letter and we will update your Data Availability statement to reflect

the information you provide.

The authors: All the sequences included in this paper have been released to the public database July 1, 2021 (https://www.ncbi.nlm.nih.gov/genbank/) and are now fully available. Accession numbers are provided in Supporting Information S2 Table. We kindly ask you to update our Data Availability Statement accordingly.

3. We note that Figure 2 in your submission contain map images which may be

copyrighted. All PLOS content is published under the Creative Commons

Attribution License (CC BY 4.0), which means that the manuscript, images,

and Supporting Information files will be freely available online, and any

third party is permitted to access, download, copy, distribute, and use

these materials in any way, even commercially, with proper attribution. For

these reasons, we cannot publish previously copyrighted maps or satellite

images created using proprietary data, such as Google software (Google

Maps, Street View, and Earth). For more information, see our copyright

guidelines: http://journals.plos.org/plosone/s/licenses-and-copyright.

We require you to either (1) present written permission from the copyright

holder to publish these figures specifically under the CC BY 4.0 license,

or (2) remove the figures from your submission:

a) You may seek permission from the original copyright holder of Figure 2

to publish the content specifically under the CC BY 4.0 license. 

We recommend that you contact the original copyright holder with the

Content Permission Form

(http://journals.plos.org/plosone/s/file?id=7c09/content-permission-form.pdf)

and the following text:

"I request permission for the open-access journal PLOS ONE to publish XXX

under the Creative Commons Attribution License (CCAL) CC BY 4.0

(http://creativecommons.org/licenses/by/4.0/). Please be aware that this

license allows unrestricted use and distribution, even commercially, by

third parties. Please reply and provide explicit written permission to

publish XXX under a CC BY license and complete the attached form."

Please upload the completed Content Permission Form or other proof of

granted permissions as an "Other" file with your submission.

In the figure caption of the copyrighted figure, please include the

following text: "Reprinted from [ref] under a CC BY license, with

permission from [name of publisher], original copyright [original copyright

year]."

b) If you are unable to obtain permission from the original copyright

holder to publish these figures under the CC BY 4.0 license or if the

copyright holder's requirements are incompatible with the CC BY 4.0

license, please either i) remove the figure or ii) supply a replacement

figure that complies with the CC BY 4.0 license. Please check copyright

information on all replacement figures and update the figure caption with

source information. If applicable, please specify in the figure caption

text when a figure is similar but not identical to the original image and

is therefore for illustrative purposes only.

The following resources for replacing copyrighted map figures may be

helpful:

USGS National Map Viewer (public domain):

http://viewer.nationalmap.gov/viewer/

The Gateway to Astronaut Photography of Earth (public domain):

http://eol.jsc.nasa.gov/sseop/clickmap/

Maps at the CIA (public domain):

https://www.cia.gov/library/publications/the-world-factbook/index.html and

https://www.cia.gov/library/publications/cia-maps-publications/index.html

USGS EROS (Earth Resources Observatory and Science (EROS) Center) (public

domain): http://eros.usgs.gov/#

The authors: We apologize for the use of this map in Figure 2. We have modified figure 2 by integrating a map already published in PLOS ONE (https://doi.org/10.1371/journal.pone.0215478). This map has been modified and vectorized under Inkscape 0.91. The original holder of this map is Pr. Are Nylund, author of the present manuscript. However, he completed the Content Permission Form. This Content Permission Form is upload as an “Other” file within this new submission. We modified the caption accordingly by adding: “The Norwegian map was reprinted from https://doi.org/10.1371/journal.pone.0215478 under a CC BY license, with permission from Are Nylund, original copyright 2019. This map is similar but not identical to the original image and is therefore for illustrative purposes only " (see Lines 210-213).

Additional Editor Comments:

Dear Dr. Lagadec,

Thank you for submitting this interesting manuscript to PloSOne. As you can

see from the reviewers' comments, they are all in favour of publishing your

findings, as I am as well, but have reservations which would need to first

be addressed. These are all experienced fish pathologists whose opinions I

value highly and I was especially happy that they all consented to review

your manuscript. That they did so is no doubt also a reflection of your own

work and that of your group.

I very much agree with their suggestions and recommendations and am

convinced that by addressing the points they have raised that this will

improve your manuscript, and inevitably counter any likely queries which

would have been raised by a critical readership.

The diversity in Tenacibaculum strains you and your colleagues have

discovered in Norwegian waters is truly impressive and I have no doubt that

groups in other parts of the world will be stimulated by your findings to

explore their own regions in greater detail. This will be key to developing

vaccines which can be widely applied to countering this threat to

aquaculture and the ensuing consequences on the environment. In this

context, the additional details and information sought by the reviewers

will be invaluable for those planning such studies.

I am very much looking forward to your response to these suggestions, just

as much as I enjoyed reading the manuscript you sent in.

Kind regards,

Lloyd Vaughan

Reviewers' comments:

Reviewer's Responses to Questions

COMMENTS TO THE AUTHOR

1. Is the manuscript technically sound, and do the data support the

conclusions?

The manuscript must describe a technically sound piece of scientific

research with data that supports the conclusions. Experiments must have

been conducted rigorously, with appropriate controls, replication, and

sample sizes. The conclusions must be drawn appropriately based on the data

presented.

Reviewer #1: Partly

Reviewer #2: Partly

Reviewer #3: Yes

Reviewer #4: Yes

2. Has the statistical analysis been performed appropriately and

rigorously?

Reviewer #1: Yes

Reviewer #2: Yes

Reviewer #3: N/A

Reviewer #4: Yes

3. Have the authors made all data underlying the findings in their

manuscript fully available?

The PLOS Data policy [2] requires authors to make all data underlying the

findings described in their manuscript fully available without restriction,

with rare exception (please refer to the Data Availability Statement in the

manuscript PDF file). The data should be provided as part of the manuscript

or its supporting information, or deposited to a public repository. For

example, in addition to summary statistics, the data points behind means,

medians and variance measures should be available. If there are

restrictions on publicly sharing data--e.g. participant privacy or use of

data from a third party--those must be specified.

Reviewer #1: No

Reviewer #2: Yes

Reviewer #3: Yes

Reviewer #4: Yes

4. Is the manuscript presented in an intelligible fashion and written in

standard English?

PLOS ONE does not copyedit accepted manuscripts, so the language in

submitted articles must be clear, correct, and unambiguous. Any

typographical or grammatical errors should be corrected at revision, so

please note any specific errors here.

Reviewer #1: Yes

Reviewer #2: Yes

Reviewer #3: Yes

Reviewer #4: Yes

5. Review Comments to the Author

Please use the space provided to explain your answers to the questions

above. You may also include additional comments for the author, including

concerns about dual publication, research ethics, or publication ethics.

(Please upload your review as an attachment if it exceeds 20,000

characters)

Reviewer #1: To better understanding the pathogen diversity underlying fish

diseases is an important prerequisite for the control of the disease. The

submitted Ms aims to examine the diversity of Tenacibaculum spp. Bacteria

in Norwegian aquaculture. These bacteria are opportunistic pathogens and

can cause ulcerative diseases. While originally, focus was T. maritimum ,

more recent research unraveled that there is much more diversity of

Tenacibaculum species in aquaculture. The present study further extends the

knowledge on the diversity of this genus, and as such it is of relevance.

Methodologically, the authors collected bacteria from marine fish farms

suffering outbreaks of ulcerative skin disease during 2017-2019. The

bacteria were isolated from diseased tissues, were cultured and then the

16S rRNA was sequenced using MLST. Personally, I am not an expert on MLST

and thus are not in a position to judge on the adequacy of the molecular

methodology. However, I have several questions on the sampling methodology:

how many fish were sampled per farm? Were only diseased or also healthy

fish/tissues sampled? 

The authors: We thank the reviewer#1 for his very constructive comments. 

Indeed, the paper in its current form lacks details and does not allow to understand what the limits of our study were. We would first clarify the sampling methodology carried out in this study: fish presenting ulcers were sampled either in the frame of a project surveying tenacibaculosis in farmed salmonids in Norway during ulcerative disease outbreaks (fish sampled by veterinarians or fish-health biologists on site during outbreaks; culturing plates and/or tissue samples were sent to our laboratory) or in the frame of other surveying projects focusing neither on tenacibaculosis nor specifically on salmon (fish sampled by fish-health biologists from our laboratory). This sampling resulted in heterogeneity in the samples (sampling during an outbreak or not, different quantity of fish sampled per sites, sampling including or not tissue/organs) but made it possible to obtain the maximum number of bacterial strains from different hosts and different localities. We clarified that point in the M&M section (see lines 77-81) and added a new table providing details on the samplings (Supporting Information S4 Table). We have also modified Table 1.

Were all isolates from one fish pooled, were isolates

from different individuals pooled, or were all isolates treated separately?

The authors: All the bacterial strains included in this study were isolated and subsequently cloned on separate culturing plates. Each strain was therefore treated separately. It is also a prerequisite to be able to genotype with the MLST technique (MLST being a concatenation of several HK gene sequences, pure bacterial clones are needed to avoid concatenation of sequences from different strains).

Were only Tenacibuculum bacteria analysed or was a broader microbial

characterization of the diseased fish performed? In particular I would be

interested to know what evidence can be provided by the authors that the

Tenacibaculum bacteria are indeed causative to the ulcerative changes. The

mere presence of a bacterial species on a diseased fish dos not necessarily

mean that this species is the etiological agent.

The authors: We fully agree that a pathogen present on a sick fish does not necessarily mean that it is the etiological agent. Although our culture process (including media) was optimized for the detection of Tenacibaculum spp., we have monitored the growth of other bacteria, especially those known to cause ulcers on marine fish (e.g., Moritella viscosa). All the bacteria isolated were submitted to a 16S rRNA sequencing, the Tenacibaculum strains being subsequently included in the genotyping. All culture plates presented a dominant growth of Tenacibaculum spp. We isolated several other bacteria, such as Vibrio spp., Pseudoalteromonas spp., Photobacterium spp., but always small colonies in minority and outcompeted by a large number of Tenacibaculum spp. colonies. 

Tenacibaculum spp. have been shown to be marine fish ulcerative pathogens in Norway for more than a decade. The dominant growth of bacteria species belonging to the genera Tenacibaculum on culture from ulcers seems to us a reasonable proof of its implication, especially because we did not detect other species likely to be the cause of the ulcers observed. However, we should clearly specify what the limits of our detection method were, and why we are fairly convinced of the implication of Tenacibaculum spp. in these ulcers (see in M&M lines 85-87, in Results lines 157-159 and in Discussion lines 264-266).

85-87

Results:

The data based in figure 1 are based on the analysis of 7 loci - what

criteria were used to select those 7 loci?

The authors: These 7 loci are part of a MultiLocus Sequence Typing scheme developed by Habib et al. (2014). They are located within protein coding genes conserved across the family Flavobacteriacea. This scheme has been since used by other teams, thus making it possible to enrich the official MLST database (https://pubmlst.org/) and to determine the DNA relatedness of the strains isolated in this study with those present in this open-access database.

To note, each of these loci are commonly used in bacterial molecular typing and had already been used in previous studies of other bacterial species and genera. 

Line 163: Since sampling details are inadequately described (see above), it

is difficult to follow the results. For instance, in line 165 its says that

ST-160 was isolated form both the skin ulcer and the head kidney of the

same fish. Was ST-160 found in other tissues of this fish as well, or were

other tissues not analysed? And was this fish the only fish with ST-160 in

the farm/the only fish with ulcer, or were more fish positive for ST-160,

and did they all show skin ulcer? This is a lot of detail, but this detail

is critical to understand the association of T. spp with ulcer.

The authors: Indeed, the current version of this manuscript lacks some details. Besides the new S4 Table with sampling details, we added pictures from the ulcers observed on the fish (Supporting Information S5 Figure). 

In addition, while the Results provide an in-depth MLST analysis, I missed

an "epidemiological" analysis: For instance, the authors investigated a

range of fish species: were certain strains/ST associated with certain host

species or was no pattern detectable? Was there an association between farm

conditions, ulcer prevalence/severity and presence of specific strains? If

this study should go beyond a mere analysis of genetic diversity but would

like to promote understanding of the disease epidemiology, more emphasis on

this type of questions is required. In the way as presented now, the ms

unfortunately does not provide insight how the results of this study would

support "future epidemic management" (line 64)

The concluding sentence of the abstract says: "Understanding their

reservoirs and transmission pathways could help to address major challenges

in connection with prophylactic measures and development of vaccines". I

fully agree with this statement, but I do not see how this Ms as presented

now would support these aims. Interestingly, the authors address these

questions in the Discussion, but unfortunately they do not apply it to

their own study.

The authors: Unfortunately, this unbalanced sampling distribution does not allow to answer some critical questions that would allow a global understanding of the disease (e.g., association host species and ST). Indeed, numbers are not sufficient to provide sufficiently robust statistical results.

In addition to continuing to build knowledge about the genetic diversity of Tenacibaculum spp. in Norway, the main result of this study is the presence of STs associated with different hosts and different localities. This partly contradicts some previous studies showing that tenacibaculosis infection arises as local epidemics and is an important result in terms of sanitary control strategy. For instance, ST-172, regularly isolated from different host species and associated with ulcers and epidemics, deserves to be considered carefully (challenge experiment, vaccine development, etc.). Not to be able to address some of the main issues of this disease of saltwater fish is a pity, and we hope that other research programmes will address them with more appropriate sampling methods.

Reviewer #2: Phylogenetic analyses of Norwegian Tenacibaculum strains

confirm high bacterial diversity and suggest circulation of ubiquitous

virulent strains, by Lagadec et al.

In this manuscript, the authors provide useful information about the

diversity of Tenacibaculum spp. in the Norwegian fish farms. Bacteria

belonging to the Tenacibaculum genus have become important opportunistic

pathogens for many different marine fishes and lately have been associated

with ulcerative diseases in farmed salmon. The authors have collected

several strains from different regions in Norway and have used MLST

analysis to map their genetic diversity.

Although the manuscript is well-written, and the information provided is

useful there are some important points that should be considered. It seems

to me that the strains used in the study have been collected by fish vets

during routine diagnostic analyses and are not part of a specific research

survey of the team. This if fully acceptable and understandable, however if

this is true, it should be specifically mentioned in the manuscript. 

I am not very sure if the media used in the isolation would favor the growth of

all possible Tenacibaculum spp. These bacteria are difficult to isolate,

and in many cases, there is overgrowth by other faster growing

opportunistic bacteria. The use of general isolation media may have

hindered the growth of other species like Tenacibaculum maritimum.

Furthermore, I am not convinced that the heamolytic activity of

Tenacibaculum strains is a valid indicator of pathogenicity. There are

several non-haemolytic bacterial pathogens which are highly virulent.

Although the authors state that the occurrence of beta haemolysis is an

indicator of pathogenicity (supported by a relevant citation), the

non-occurrence does not mean that the strains are less virulent. Haemolytic activity has been used extensively by the authors, however how it was

assessed is not described in the M&M. More care should be given in the use

of haemolysis as virulence factor for Tenacibaculum in order to be

convincing.

The authors: We deeply thank the reviewer#2 for his pertinent remarks. 

To clarify a first point, it is important to specify that the fish were sampled either in the frame of a project surveying tenacibaculosis in farmed salmonids in Norway (fish sampled by veterinarians or fish-health biologists on site during outbreaks) or during other projects (fish sampled by certified fish-health biologists from our laboratory). We have clarified this point in the M&M section (see lines 77-81).

We answered the other questions in the specific comments below. 

Specific comments to the authors:

L53: move (MA) after Marine Agar

The authors: corrected (see line 53).

L53: Explain how the use of MA associates Tenacibaculum with winter ulcers

The authors: Tenacibaculum spp. are indeed difficult to isolate. Originally, blood agar containing 1.5-2.0 % NaCl (BAS) was used as the medium for investigation of bacterial diseases in saltwater salmon life stages. The use of Marine Agar (MA) which includes sea salts, has improved the isolation of Tenacibaculum bacteria. Several papers mention that some Tenacibaculum spp. only grow in the presence of sea salts [1-3]. We have added references in the Introduction and modified the sentence accordingly (see lines 52-54).

L68: Mention how many samples were used

The authors: We added this information in the M&M. The sentence “A total of 197 fish […]” was added line 88. The number of fish per sampling is now available in a new table (Supporting Information S4 Table).

L75-78: Are these the best media for isolating Tenacibaculum spp? What

about FMA or TMA? It seems that these media are mostly used for routine

microbiology by the fish vets. If this is the case, it should be stated

either in the M&M or in the introduction, since this is a limitation of the

study that may have biased the results. If this is not the case, then you

should provide references supporting the use of these media for the

specific isolation of Tenacibaculum spp.

The authors: Marine Agar (MA) is indeed used routinely by fish veterinarians. We specified this point in the M&M, line 83.

To prevent the outgrowth by faster growing bacteria, we also used a blood marine agar supplemented with kanamycin to a final concentration of 50 µg ml-1 (KABAMA). Based on our experience, this medium has drastically improved the recovery of Tenacibaculum spp.

L:78 It is not clear later in the results how many of the isolates were

Tenacibaculum. If there were other bacteria present, which were these? What

is the prevalence of Tenacibaculum during the outbreaks?

The authors: All the isolates specified in this paper are referring to Tenacibaculum spp. isolates. 

This point was also raised by reviewer#1. We answered to his comment:

“Although our culture process (including media) was optimized for the detection of Tenacibaculum spp., we have monitored the growth of other bacteria, especially those known to cause ulcers on marine fish (e.g., Moritella viscosa). All the bacteria isolated were submitted to a 16S rRNA sequencing, the Tenacibaculum strains being subsequently included in the genotyping. All culture plates presented a dominant growth of Tenacibaculum spp. We isolated several other bacteria, such as Vibrio spp., Pseudoalteromonas spp., Photobacterium spp., but always small colonies in minority and outcompeted by a large number of Tenacibaculum spp. colonies. 

Tenacibaculum spp. has been showed to be a marine fish ulcerative pathogens in Norway for more than a decade. The dominant growth of bacteria species belonging to the genera Tenacibaculum on culture from ulcers seems to us a reasonable proof of its implication, especially because we did not detect other species likely to be at the origin of the ulcers observed. However, we should clearly specify what the limits of our detection method were, and why we are fairly convinced of the implication of Tenacibaculum spp. in these ulcers (see in M&M lines 85-87, in Results lines 157-159 and in Discussion lines 264-266).”

L80: What are the characteristics of the colonies? are these the same in

all media? were they described also in Blood Agar? In the papers cited the

authors have used different media.

The authors: Colonies characteristics are provided in a new S3 Table. These colonies were described after growth on BAMA medium.

L82: Describe the morphology

The authors: We modified the sentence, which now reads: “The colonies were further examined by using a light microscope to identify the long and slender rods typical of Tenacibaculum spp. cell morphology” (lines 91-92).

L143: Table legend: You talk about haemolytic activity but how this was

assessed is not in the M&M. Since haemolytic capacity of the bacteria plays

an important role in your paper, the method used to assess it should be

described

The authors: This is an important point. We added a sentence in the M&M to detail how we assessed the haemolytic activity of the different strains: “The β-haemolytic activity of the colonies was readily observed by looking at the plates through a light source.” (see lines 92-94).

L190-191: How do you define "variable"? Have you used replicates in the

heamolytic activity assay?

The authors: The only “variable” strain in this study is the newly described T. finnmarkense genomovar finnmarkense [4]. Its haemolytic activity has been qualified as “variable”. They tested haemolysin production on Marine Agar with 5% bovine blood and 50 µg ml−1 kanamycin and recorded within 7 days. They performed phenotypic tests in parallel.

We assessed the haemolytic properties of our strains in duplicates and did not notice any variable haemolytic activity of the other strains.

L266: What are the severe symptoms? Where were they described?

The authors: Outbreaks characteristics and sampling details are described in S4 table. We also added a new S5 Figure with pictures of the ulcers for 9 different samplings (including Sampling 28).

L276: How did you come to this conclusion that the non-haemolytic strains

are less pathogenic? If the conclusion is based on the field data (eg

morbidity/mortality), these data should be provided. Even then, the

conclusion is really based on circumstantial evidence, and it is indicative

if not speculative. Unless there are comparative data from challenge tests

(if yes, then you should provide these data).

L281: The association of hemolysis to virulence or to be more accurate, the

non-hemolysis to non- or reduced virulence is not supported by evidence and

it is speculative.

The authors: We agree that precautions should be taken when discussing the link between haemolytic activity and pathogenicity, especially concerning the virulence of the non- haemolytic bacteria. We have modified the paragraph accordingly (see lines 307-311).

Reviewer #3: This manuscript provides additional information on the

epidemiology of Tenacibaculosis and should be published. The information

provided is sound and useful but not a significantly novel contribution to

fish bacteriology, but should be published as it provides good information

on the distribution and epidemiology of the bacterial isolates of concern.

The authors: We deeply thank the Reviewer #3. It is true that this paper does not add significant new insights to fish bacteriology, but we think these new findings are relevant for a better understanding of Tenacibaculum spp. diversity and tenacibaculosis.

Reviewer #4: The manuscript submitted by Lagadec et al (PONE-D-21-21158) is

a well presented and informative paper showing the diversity that exists

amongst Tenacibaculum spp. strains in Norwegian aquaculture.

I have a few minor points for the authors to address.

The authors: We thank Reviewer#4 for his comments. Please find our answers to each specific points below.

Line 39 - bacterial diseases

The authors: corrected (see line 39).

Line 44 - Tenacibaculosis has been reported in marine wild and farmed fish

in Europe, Asia,

The authors: corrected (see line 44).

Line 52 - Due to the increased use of Marine Agar (MA) for bacterial

isolation in recent years, Tenacibaculum spp. have also been associated

with the winter ulcer disease, previously attributed to the bacterium

Moritella viscosa [25, 26].

The authors: corrected (see lines 52-54).

Line 89 - 10,000 rpm - state xg rather than rpm

The authors: That was a mistake. It was corrected (see line 101).

Lines 139/194/213/294 - Numbers at the start of sentences should be written

in words rather than as a number.

The authors: We apologize for these mistakes. We have corrected these sentences accordingly (see lines 151, 215, 234, 323).

Line 151- March 2019

The authors: corrected (see line 168).

Line 161 - Of the 27 new STs, 22 were isolated during a single outbreak of

tenacibaculosis. Was the outbreak at a single site - mention the location

of this outbreak?

The authors: Our sentence was confusing. Twenty-two new STs were isolated only once, but during 22 different outbreaks. We have modified this sentence, which now reads “Of the 27 new STs, 22 were isolated only once during 22 different outbreaks of tenacibaculosis” (line 178). We have also modified a sentence in the Discussion (see lines 324-326).

Line 278 - Overall, non-haemolytic Tenacibaculum strains seem to be less

pathogenic and may not lead to tenacibaculosis outbreaks. Can you state the

evidence for this e.g. a reference or findings from your study?

The authors: This point has been raised by Reviewer#2. We should have taken more precautious in writing this paragraph. Indeed, even if the non-haemolytic strains in our study seem to be less virulent, some STs (e.g. ST-160) were associated with tenacibaculosis. We have modified the paragraph accordingly (see lines 307-311).

6. PLOS authors have the option to publish the peer review history of their

article (what does this mean? [3]). If published, this will include your

full peer review and any attached files.

If you choose "no", your identity will remain anonymous but your review may

still be made public.

DO YOU WANT YOUR IDENTITY TO BE PUBLIC FOR THIS PEER REVIEW? For

information about this choice, including consent withdrawal, please see our

Privacy Policy [4].

Reviewer #1: No

Reviewer #2: No

Reviewer #3: No

Reviewer #4: No

While revising your submission, please upload your figure files to the

Preflight Analysis and Conversion Engine (PACE) digital diagnostic tool,

https://pacev2.apexcovantage.com/. PACE helps ensure that figures meet PLOS

requirements. To use PACE, you must first register as a user. Registration

is free. Then, login and navigate to the UPLOAD tab, where you will find

detailed instructions on how to use the tool. If you encounter any issues

or have any questions when using PACE, please email PLOS at

figures@plos.org. Please note that Supporting Information files do not need

this step.

1. Suzuki M, Nakagawa Y, Harayama S, Yamamoto S. Phylogenetic analysis and taxonomic study of marine Cytophaga-like bacteria: proposal for Tenacibaculum gen. nov. with Tenacibaculum maritimum comb. nov. and Tenacibaculum ovolyticum comb. nov., and description of Tenacibaculum mesophilum sp. nov. and Tenacibaculum amylolyticum sp. nov. Int J Syst Evol Microbiol. 2001;51(Pt 5):1639-52. Epub 2001/10/12. doi: 10.1099/00207713-51-5-1639. PubMed PMID: 11594591.

2. Olsen AB, Nilsen H, Sandlund N, Mikkelsen H, Sorum H, Colquhoun DJ. Tenacibaculum sp. associated with winter ulcers in sea-reared Atlantic salmon Salmo salar. Dis Aquat Organ. 2011;94(3):189-99. Epub 2011/07/28. doi: 10.3354/dao02324. PubMed PMID: 21790066.

3. Småge SB, Brevik OJ, Duesund H, Ottem KF, Watanabe K, Nylund A. Tenacibaculum finnmarkense sp. nov., a fish pathogenic bacterium of the family Flavobacteriaceae isolated from Atlantic salmon. Antonie Van Leeuwenhoek. 2016;109(2):273-85. Epub 2015/12/15. doi: 10.1007/s10482-015-0630-0. PubMed PMID: 26662517; PubMed Central PMCID: PMCPMC4751178.

4. Olsen AB, Spilsberg B, Nilsen HK, Lagesen K, Gulla S, Avendaño-Herrera R, et al. Tenacibaculum piscium sp. nov., isolated from skin ulcers of sea-farmed fish, and description of Tenacibaculum finnmarkense sp. nov. with subdivision into genomovars finnmarkense and ulcerans. Int J Syst Evol Microbiol. 2020. doi: https://doi.org/10.1099/ijsem.0.004501.

---

## [Editor Report · Decision Letter 1]

15 Oct 2021

Phylogenetic analyses of Norwegian Tenacibaculum strains confirm high bacterial diversity and suggest circulation of ubiquitous virulent strains

PONE-D-21-21158R1

Dear Dr. Lagadec,

We’re pleased to inform you that your manuscript has been judged scientifically suitable for publication and will be formally accepted for publication once it meets all outstanding technical requirements.

Kind regards,

Lloyd Vaughan, PhD, BSc (Hons)

Academic Editor

PLOS ONE
---

## [Editor Report · Acceptance letter]

19 Oct 2021

PONE-D-21-21158R1 

Phylogenetic analyses of Norwegian *Tenacibaculum* strains confirm high bacterial diversity and suggest circulation of ubiquitous virulent strains. 

Dear Dr. Lagadec:

I'm pleased to inform you that your manuscript has been deemed suitable for publication in PLOS ONE. Congratulations! Your manuscript is now with our production department. 

Kind regards, 

on behalf of

Prof. Dr. Lloyd Vaughan 

Academic Editor

PLOS ONE